# Wet-dry cycles enable the parallel origin of canonical and non-canonical nucleosides by continuous synthesis

Sidney Becker [1], Christina Schneider [1], Hidenori Okamura[1], Antony Crisp [1], Tynchtyk Amatov [1], Milan Dejmek[2] & Thomas Carell [1]

The molecules of life were created by a continuous physicochemical process on an early Earth. In this hadean environment, chemical transformations were driven by fluctuations of the naturally given physical parameters established for example by wet–dry cycles. These conditions might have allowed for the formation of (self)-replicating RNA as the fundamental biopolymer during chemical evolution. The question of how a complex multistep chemical synthesis of RNA building blocks was possible in such an environment remains unanswered. Here we report that geothermal fields could provide the right setup for establishing wet–dry cycles that allow for the synthesis of RNA nucleosides by continuous synthesis. Our model provides both the canonical and many ubiquitous non-canonical purine nucleosides in parallel by simple changes of physical parameters such as temperature, pH and concentration. The data show that modified nucleosides were potentially formed as competitor molecules. They could in this sense be considered as molecular fossils.

[1] Center for Integrated Protein Science Munich CiPSM at the Department of Chemistry, Ludwig-Maximilians-Universität München, 81377 Munich, Germany.
[2] Institute of Organic Chemistry and Biochemistry ASCR, 16610 Prague 6, Czech Republic. Correspondence and requests for materials should be addressed to T.C. (email: thomas.carell@lmu.de)

The molecules of life originated around 4 billion years ago under conditions governed by the composition of the Earth's crust and atmosphere at that time[1, 2]. Molecules such as nucleic acids and amino acids must have formed by a continuous physicochemical process, in which greater structural complexity was generated based on fluctuations of the naturally given physical parameters[3]. Geothermal fields, for example, could have established such fluctuations by wet–dry cycles that may have driven chemical transformations, which ultimately allowed the emergence of life[4–8]. The appearance of (self)-replicating RNA was certainly of central importance for the transition from an abiotic world to biology[9–11]. We need to consider, however, that an early genetic polymer might have been structurally different from contemporary RNA. This involves differences regarding the sugar configuration (e.g., pyranosyl RNA) or the presence of other nucleobases[12, 13]. Selection pressure led in this scenario to the chemical evolution of RNA. Contemporary RNA molecules contain four canonical nucleosides (A, G, C, U), which establish the sequence information. In addition, >120 non-canonical nucleosides are present, which govern a diverse set of properties such as correct folding, e.g., to enable catalysis[14]. In fact, the genetic system of all known life is dependent on modified nucleosides. Many of these non-canonical nucleosides are found today in all three domains of life, which indicates that they were present early on during the development of life. For the ubiquitous non-canonical nucleosides we may assume that they were already formed as competitors in parallel with the canonical ones on the early Earth[15]. So far, however, a geochemical scenario that would allow for the parallel formation of canonical and non-canonical RNA building blocks by a continuous process is not known. All reported multistep chemical models so far rely on tightly controlled laboratory conditions and the isolation and purification of central reaction intermediates by sophisticated methods[1, 16–19].

Herein, we report a robust synthetic pathway, which is purely based on fluctuations of physicochemical parameters such as pH, concentration, and temperature, driven by wet–dry cycles. These fluctuations enable the direct enrichment or purification of all reaction intermediates that are directly used for the next synthetic steps. As such, a continuous synthesis is established. Our results show that RNA building blocks can indeed be formed in a prebiotically plausible geochemical environment without sophisticated isolation and purification procedures. The chemical scenario presented here supports the hypothesis that life may have originated in a hydrothermal milieu on land rather than in a deep sea environment. The key assembly step in our pathway is the formation of variously substituted 5-nitroso-pyrimidines (nitrosoPys) that can be converted into formamidopyrimidines (FaPys) in the presence of formic acid and elementary metals (Ni or Fe). When combined with ribose, the FaPy compounds react to give a set of purine nucleosides. This chemical pathway delivers not only the canonical purine nucleosides but concomitantly many of the ubiquitously present non-canonical relatives, suggesting their origin as prebiotic competitor nucleosides (A, ms$^2$A, m$^2$A, DA, G, m$^2$G, m$^2_2$G, m$^1$G). Since chemical evolution depended on those molecules that were available on early Earth, these non-canonical nucleosides may be considered to be molecular fossils, which maintained their essential life-supporting character until the present day.

## Results

**Selective crystallization of an organic salt**. The chemical scenario that leads to a continuous synthesis of RNA building blocks by just fluctuations of physical parameters is shown in Figs. 1 and 2a. The scenario starts with an aqueous solution of malononitrile 1 and different amidinium salts 2a-d (HCl or H$_2$SO$_4$ salts, 400 mM), both recognized prebiotic compounds[18]. In addition,

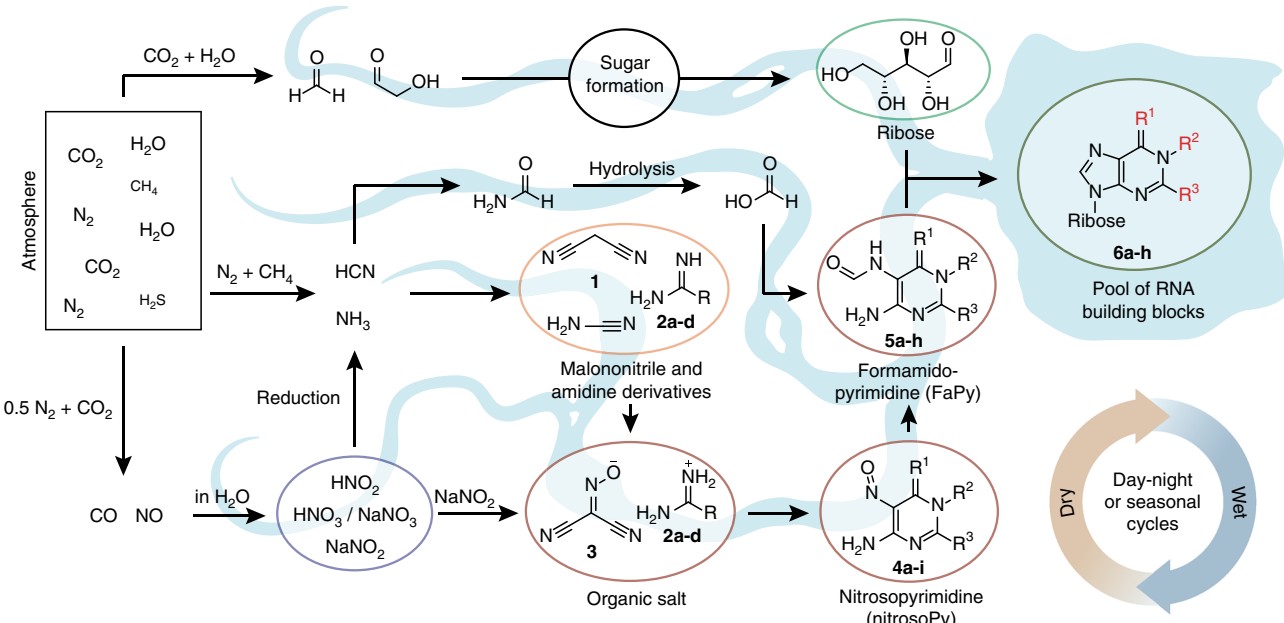

**Fig. 1** RNA nucleoside formation pathway. A geothermal environment provides the right set up for the depicted transformations by establishing wet–dry cycles. The prebiotic starting materials are produced from a prebiotic atmosphere and washed into an aqueous environment (e.g. by rain). Major atmospheric components are written in larger letters, whereas minor components are written in smaller letters. Transformations are taking place in different environments, illustrated by various rivers (in light blue). Each environment provides the right setup for different chemistries, leading to several different chemical transformations. This geochemical setup leads to a set of canonical and non-canonical RNA building blocks by continuous synthesis (6a, m$^1$G: R$^1$ = O, R$^2$ = Me, R$^3$ = NH$_2$; 6b, ms$^2$A: R$^1$ = NH, R$^2$ = H, R$^3$ = SMe; 6c, A: R$^1$ = NH, R$^2$ = H, R$^3$ = H; 6d, m$^2$G: R$^1$ = O, R$^2$ = H, R$^3$ = NHMe; 6e, m$^2_2$G: R$^1$ = O, R$^2$ = H, R$^3$ = N(Me)$_2$; 6f, G: R$^1$ = O, R$^2$ = H, R$^3$ = NH$_2$; 6g, DA: R$^1$ = NH, R$^2$ = H, R$^3$ = NH$_2$; 6h, m$^2$A: R$^1$ = NH, R$^2$ = H, R$^3$ = Me)

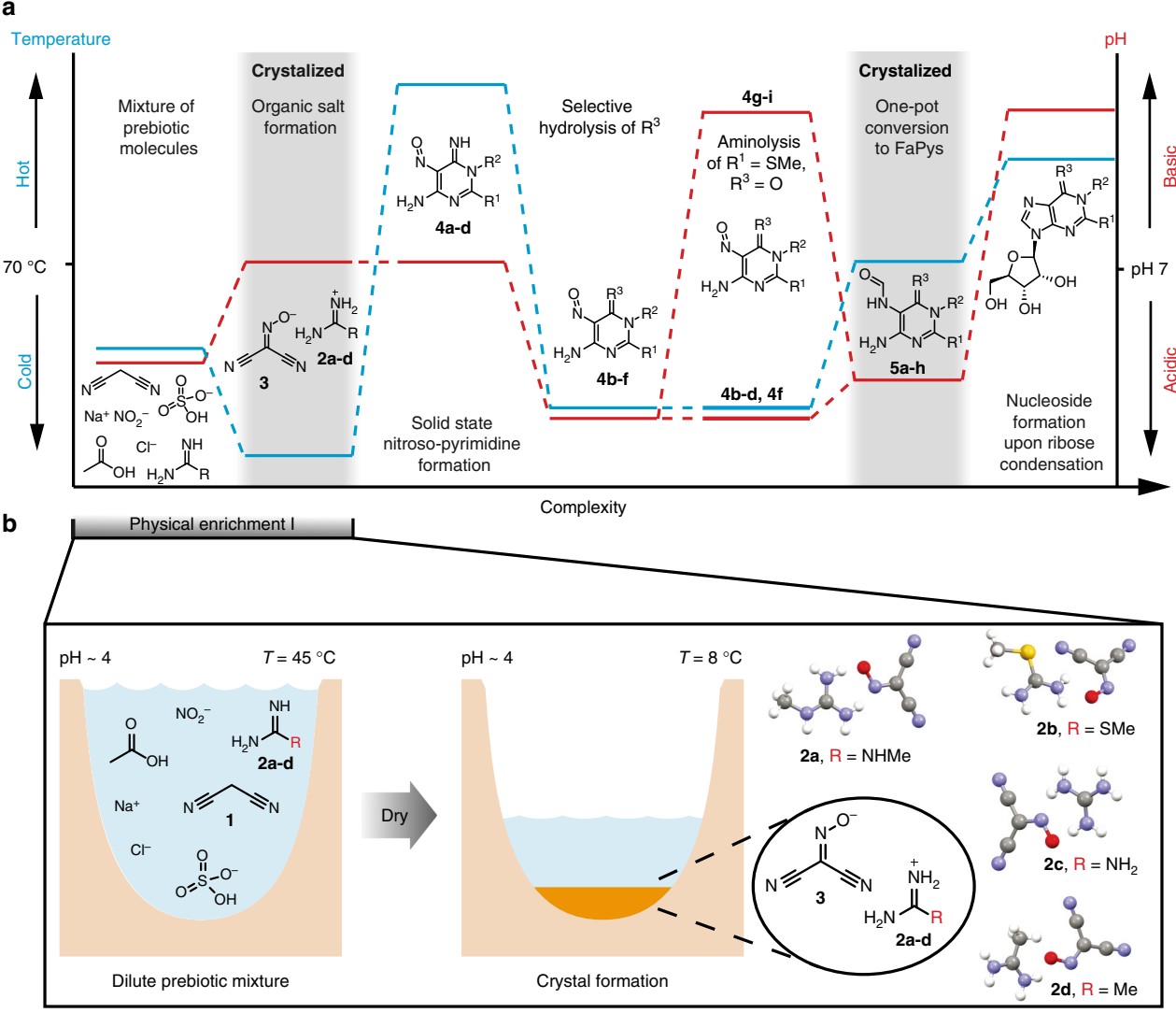

**Fig. 2** Chemical complexity created by physical fluctuations. **a** Relative changes of temperature (in blue) and pH (in red) are shown for each synthetic step for the continuous synthesis of purine RNA building blocks from small organic and inorganic molecules. Several wet–dry cycles establish fluctuations of the depicted physical parameters that enable the physical enrichment of intermediates. Gray backgrounds denote compounds that are enriched by crystallization from an aqueous solution. **b** Formation of an organic salt consisting of amidine derivatives **2a-d** and (hydroxyimino)malononitrile **3**. The salt is selectively crystalized by concentrating a dilute mixture of organic and inorganic compounds by slow evaporation. The crystal structures of the four crystalized organic salts are depicted (Supplementary Tables 1–4)

sodium nitrite and acetic acid are present to establish a slightly acidic pH of around 4. Under these conditions the amidine molecules (**2a-d**) are protonated, which leads to their chemical deactivation. This allows selective nitrosation of malononitrile **1** to give (hydroxyimino)malononitrile **3** in situ. Slow evaporation of water under ambient conditions, followed by gentle cooling to 8–10 °C resulted in crystallization of a salt from the ca. 1 M amidinium solution. This crystallization is very robust and resembles naturally occurring concentration processes. The resultant crystals had excellent quality for X-ray analysis, which showed that the salts are formed from the amidinium cations **2a-d** and the (hydroxyimino)malononitrile anion of **3** (Fig. 2b). Interesting is the distance between the negatively charged oxygen in **3** and the positively charged H-bond donor centre of the amidininium units **2a-d**. We determined distances between 1.85–1.95 Å, which is long for a salt bridge but right in the regime for a typical hydrogen bond. This is important because it is supposedly the reason for the comparably low melting temperatures of the salts, which we determined between 110 and 160 °C.

The robustness and ease of crystallization establishes a first physical enrichment step that finishes the initial wet–dry phase with the deposition of these salt materials (Fig. 2b).

**Nitroso-pyrimidine formation**. When the obtained salts containing **2a-d** and **3** are subsequently heated to their respective melting temperatures, transformation into the corresponding nitroso-pyrimidines (**4a-d**, Fig. 3a) occurs. The required temperatures between 110 and 160 °C could have been readily accessible under early Earth conditions, due to, for example, volcanic activity in geothermal fields or sunlight shining on dark surfaces. In order to investigate whether the nitroso-compounds **4a-d** would form in parallel despite their varying structures and different melting points, the different salts were combined in a reaction flask and a temperature gradient (1 °C/5 min, from 100–160 °C) was applied to simulate soil that would slowly heat up. Subsequent $^1$H-NMR analysis indicated successful formation of the anticipated nitroso-pyrimidines **4a-d** (Supplementary Fig. 1).

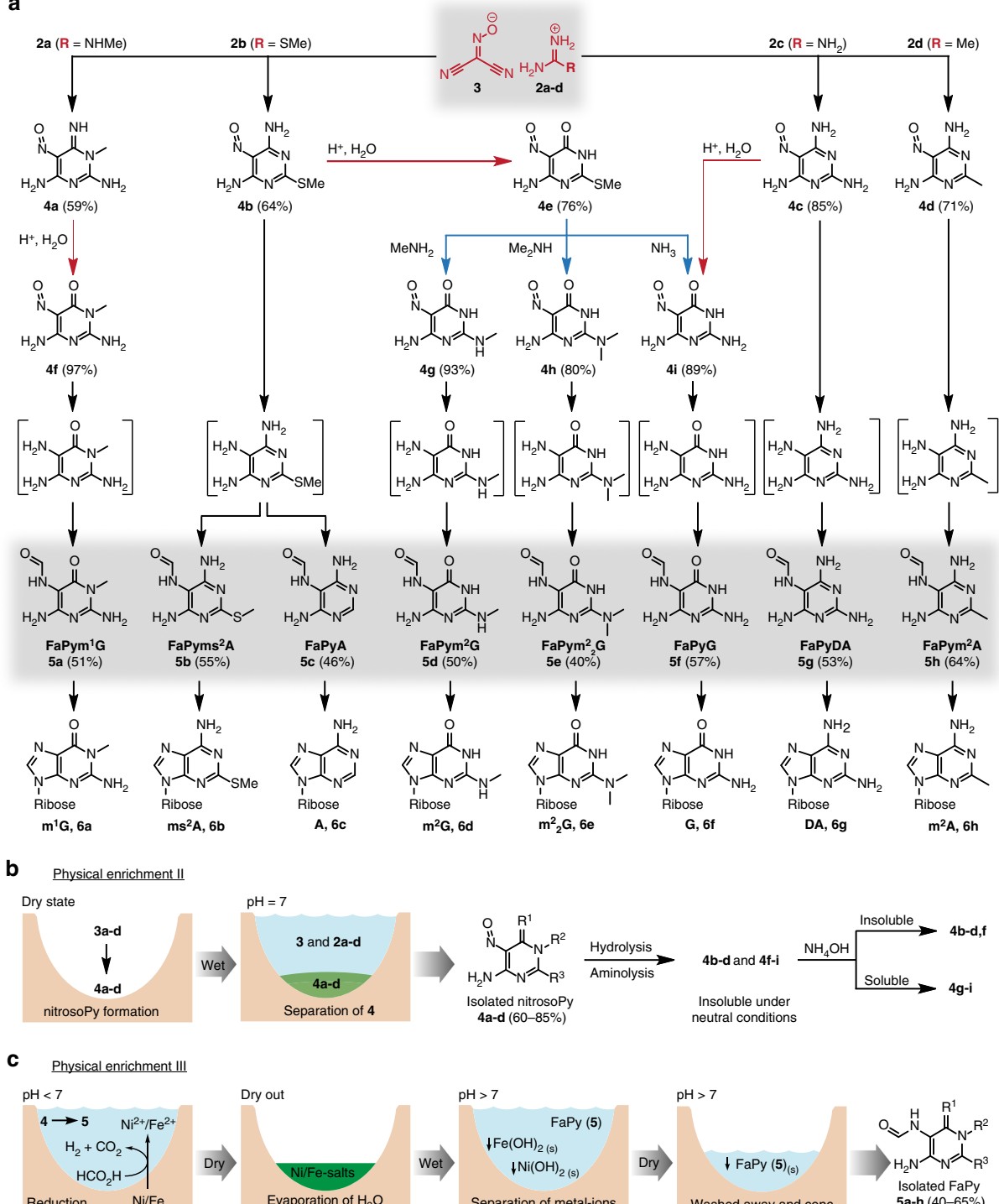

**Fig. 3** Reaction scheme and physical enrichment of intermediates. **a** Dry-state reactions of salts containing **2a-d** and **3** provide nitroso-pyrimidines **4a-d**, which can be further diversified by hydrolysis (red arrows) or aminolysis (blue arrows) to give a set of nitroso-pyrimidines (nitrosoPys) **4a-i**. In the presence of elementary Fe and Ni and dilute formic acid, formation of the formamidopyrimidines (FaPys) **5a-h** as direct purine base precursors takes place. In square brackets: non-isolated reaction intermediates. **b** Second physical enrichment of the nitroso-pyrimidines isolated in high purity and yield. **c** Third physical enrichment of the formed FaPys **5a-h** as nucleoside precursors from nitroso-pyrimidines

The resultant nitroso-pyrimidines are stable compounds with melting points typically >250 °C without decomposition. In addition we noted that the nitroso-pyrimidines are rather insoluble in water, which offers the possibility for a second physical enrichment step. Addition of water to the reaction mixture dissolves unreacted starting materials, leaving the nitroso-pyrimidines in basically NMR-pure form behind (Supplementary Fig. 2). In this model, one wet–dry cycle and two physical enrichment steps with a final rain shower or flooding would be sufficient to deposit a mixture of stable nitroso-pyrimidines (**4a-d**) in excellent purities and good chemical yields between 60 and 85% (Fig. 3a).

**Diversification by hydrolysis and aminolysis**. Depending on the composition and pH of the aqueous environment, which may or may not contain different amines, the nitroso-pyrimidines could undergo further hydrolysis and aminolysis reactions (Fig. 3a). Because these reactions are very slow under neutral conditions, we used dilute HCl to accelerate the processes for investigation. Importantly, we noted a high regioselectivity. Upon treatment overnight at room temperature with 0.5 M HCl, compounds **4a** and **4c** for example are hydrolyzed to afford the oxo-nitroso-pyrimidines **4f** and **4i** in near quantitative yields. Hydrolysis of **4b** to product **4e** was comparitively slower, and under our accelerated conditions a mixture of **4b** and **4e** was obtained. This inefficient conversion would be advantageous in a prebiotic context given that from **4b** the canonical nucleoside adenosine (A) and its 2-thiomethyl derivative (ms$^2$A) are derived later, whereas **4e** gives rise to guanosine derivatives (G, m$^2$G, m$^2_2$G, Fig. 3a). This allows for the simultaneous formation of canonical and non-canonical bases from the same precursor. In contrast to the 2-amino (**4a,c**) or 2-methyl (**4d**) substituted nitroso-pyrimidines, we noted that the 2-thiomethyl functionality in **4b** and **4e** was prone to undergo selective nucleophilic substitution. Reaction of **4e** with different amines leads to efficient formation of the nitroso-pyrimidines **4g-i** with the concomitant release of methanethiol. Due to its insolubility under basic conditions, nucleophilic substitutions of **4b** are very inefficient. To confirm this, we partially hydrolyzed **4b** to **4e** in the presence of methylamine (300 mM) and dimethylamine (100 mM). The pH was carefully adjusted with Na$_2$CO$_3$ to about pH 10. Compound **4b** precipitated, while **4e** stayed in solution, consequently protecting **4b** from further reactions. It is in this context interesting that nucleosides that would form via aminolysis of **4b** have not yet been found in nature. In contrast, **4e** reacts efficiently and after 3–4 days at room temperature **4e** is almost completely converted into **4g** and **4h**, which are direct precursors to the ubiquitous non-canonical RNA bases m$^2$G and m$^2_2$G (Fig. 3a, Supplementary Fig. 3).

Thus, a few simple chemoselective and regioselective hydrolysis and aminolysis reactions affords a diverse mixture of differently substituted nitroso-pyrimidines (**4b-d, f-i**), all of which possess the right substitution pattern for the synthesis of naturally occuring canonical and non-canonical RNA nucleosides. Because all the formed nitroso-pyrimidines are poorly soluble in water at neutral pH, neutralizing the solutions leads to their efficient precipitation, providing a naturally occurring purification step (Fig. 3b). Importantly, all nitroso-pyrimidines that later give adenosine-derived nucleosides (**4b-d**) are soluble in water under acidic conditions, while the nitroso-compounds that are converted into guanosine-derived nucleosides (**4g-i**, except for **4f**) are soluble under basic pH conditions. These properties allow for potentially divergent chemical pathways leading to A-derived and G-derived nucleosides (Fig. 3b, Supplementary Fig. 4).

**Formamidopyrimidine formation as nucleobase precursor**. The next wet–dry cycles allow for the formation and isolation of formamidopyrimidines (FaPys) **5a-h**, from nitroso-pyrimidines **4** that are after their precipitation exposed to acidic conditions like dilute formic acid in the presence of elementary Fe or Ni, which are components of the Earth's crust. This leads to reduction of the nitroso-pyrimidines **4** to aminopyrimidines as non-isolated reaction intermediates (Fig. 3a, in square brackets), which are immediately formylated to give the water soluble formamidopyrimidines (FaPys) **5a-h** in a one-pot reaction. During the wet phase, Ni$^0$ and Fe$^0$ are converted into the biologically relevant Ni$^{2+}$/Fe$^{2+}$ ions, while formic acid decomposes into CO$_2$ and H$_2$ (Fig. 3c). In the reaction formic acid has a dual function. It provides the H-atoms needed for the reduction and it

subsequently reacts with the formed aminopyrimidines to give FaPy compounds that were already shown to be prebiotically valid precursors to purine nucleosides[18]. The Ni/Fe/formic acid environment converts quantitatively all nitroso-compounds **4b-d, f-i** into the corresponding FaPy compounds **5a-h** (Fig. 3a). The water soluble FaPy compounds (under dilute basic conditions) can now be separated from unreacted Ni$^0$/Fe$^0$ and from the formed Ni$^{2+}$/Fe$^{2+}$ byproducts. Under slightly basic conditions (pH ≈ 9–10) the latter compounds precipitate as insoluble carbonate or hydroxide salts. The FaPys **5a-h** are thus washed away, while the transition metal compounds sediment out. Final evaporation of water concentrates the reaction mixture, leading to the crystallization of the FaPy molecules. This third physical enrichment step, involving a wet–dry cycle, leads to the NMR-clean formation of FaPy-derivatives **5a-h** (Fig. 3c).

The 2-(methylthio)-5-nitrosopyrimidine-4,6-diamine (**4b**) gives after treatment with formic acid and elementary Ni two different FaPy products depending on the reaction conditions. One of the products (**5b**) contains a thiomethyl group, while the other (**5c**) is desulfurated. The desulfurization reaction is simply controlled by time and can be promoted when H$_2$ is bubbled through the solution prior to reaction. Compound **5c** is always generated in a stepwise reaction cascade via compound **5b** which was confirmed by reacting **4b** for 2 h and isolating the only product formed (**5b**, Fig. 3c). The isolated product was immediately subjected to the same conditions, which provided **5c** after 7 days in pure form. This pathway via nitroso-pyrimidines thus affords **5c**, the precursor for the canonical base A under plausible prebiotic conditions[20]. These conditions also lead to the parallel formation of the precursor to the ubiquitous 2-thiomethyl modification (ms$^2$A), which is today found in all three domains of life.

**Formation of canonical and non-canonical nucleosides**. All of the prepared FaPy compounds undergo rapid and regioselective condensations with ribose when they are present in the same dry-state environment (Fig. 4). We do not assume that ribose was formed at the same location together with the FaPy compounds since the required carbohydrate chemistry may be incompatible. Several models are available, however, that show ribose formation in different physical environments[21–24]. Even though ribose and FaPys might have formed separately, the water solubility of the FaPys and of ribose allows them to be washed into the same environment by rain or flooding. Evaporation of water in the last wet–dry cycle would enable a condensation reaction under dry-state conditions. Indeed, the physically enriched FaPy compounds (**5a-h**) engage in a rapid reaction with ribose to give the corresponding FaPy-ribosides. Upon dissolution in water and subsequent heating under basic conditions, all four expected purine α/β-ribofuranosides (α/β–f) and α/β-pyranosides (α/β–p) are obtained (**6a-h**, Fig. 4a), completing the last wet–dry cycle. The LC-MS traces of the reactions using both UV- and MS-detection are shown in Fig. 4b. To ensure correct structural assignment we chemically synthesized some of the expected products and performed co-injection studies (Supplementary Methods). These experiments show that the major isomers are the naturally occurring β-configured pyranosides and furanosides. Pyranosides are building blocks for pyranosyl-RNA, which was suggested to be a potential RNA predecessor[12]. Therefore, our scenario delivers the building blocks for this pre-RNA and for RNA. As such it provides the basis for the chemical transition from one genetic polymer to the other directed by selection pressure. Importantly, our continuous synthetic pathway provides next to the canonical bases A and G also the non-canonical β-furanosyl-nucleosides (β–f) m$^2$G, m$^2_2$G, m$^1$G, ms$^2$A and m$^2$A (in red, Fig. 4b), arguing

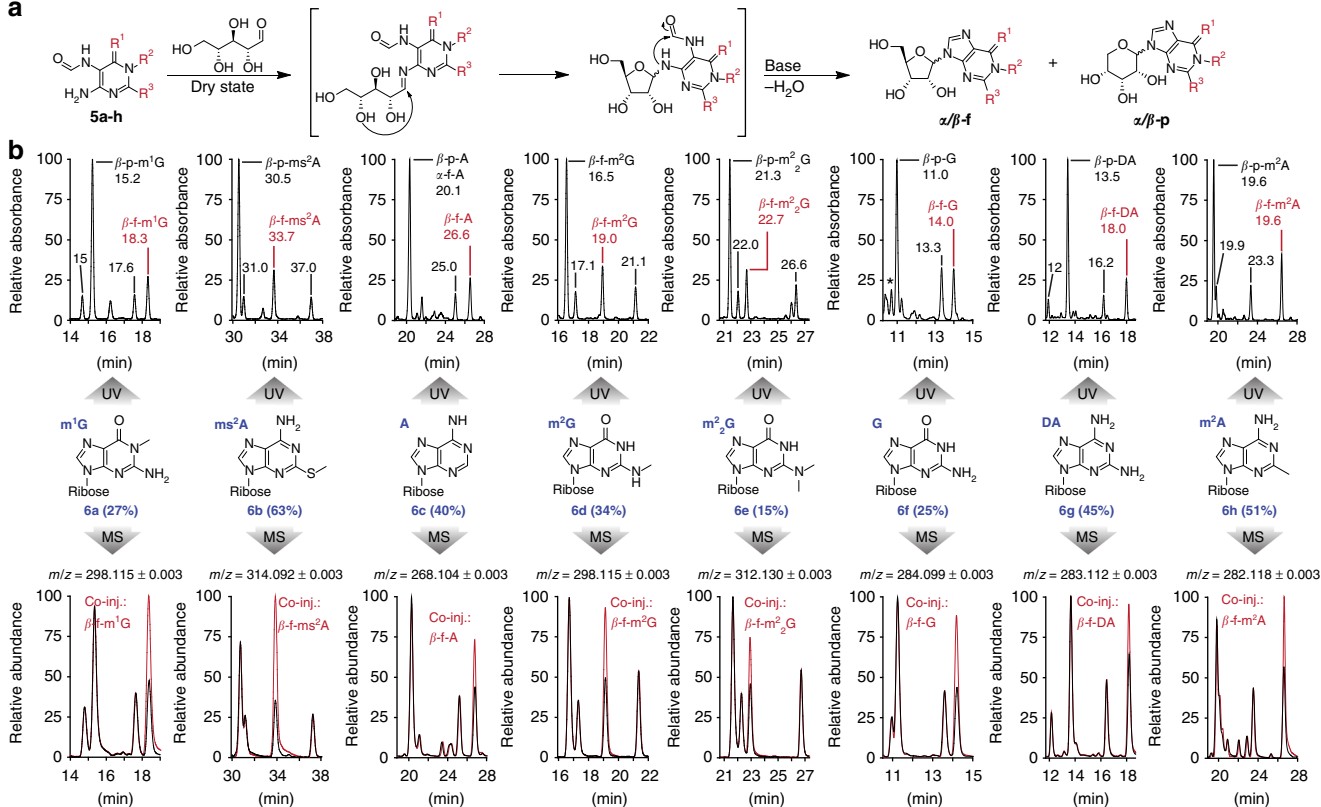

**Fig. 4** Formation of RNA nucleosides from nucleobase precursors. **a** Reaction mechanism for the formation of canonical and non-canonical RNA nucleosides **6a-h** from formamidopyrimidines (FaPys) **5a-h** and ribose. The reaction provides the four expected isomers $\alpha/\beta$ ribopyranosides ($\alpha/\beta$−p) and $\alpha/\beta$-ribofuranosides ($\alpha/\beta$−f). **b** LC-MS analysis of the reaction products from **5a-h** and ribose. Compounds identified by MS detection are labeled ($\beta$−p or $\alpha/\beta$−f). The UV- and MS-chromatograms show all four expected isomers for each compound (labeled with the retention time or asterisk (*) in the UV-chromatogram). The structural assignment was assisted by co-injection (Co-inj.) studies. The MS traces show in red the co-injection signal obtained with the naturally occurring isomer ($\beta$−f)

that the early RNA polymer was structurally already more complex regarding the nucleobases. The ribosylation of the FaPys leading to non-canonical nucleosides is equally efficient to the formation of A and G, with yields between 15 and 60%. Interestingly, we noted that for some A derivatives (m²A, DA and A) other regioisomers were found as well. These isomers were not formed when pure FaPy starting materials were used that were not derived from our continuous synthesis. We believe that these isomers might be the N3-connected nucleosides, previously proposed by Wächtershäuser for homo-purine RNA[25]. Despite the presence of these side products, we observe efficient N9-nucleoside formation with remarkable yields of up to 60% for the canonical and the non-canonical nucleosides. This work demonstrates that the non-canonical compounds could plausibly have formed as companion and potential competitor compounds in parallel to the canonical nucleosides.

## Discussion

Life on earth certainly did not start in a chemists' laboratory, where the relevant compounds are assembled in a step-by-step process from pure starting materials under tightly controlled conditions. Even if the individual reaction steps are performed under plausibly prebiotic conditions, the controlled assembly over many steps with sophisticated isolation and purification procedures of reaction intermediates is an unlikely scenario for chemical synthesis under early Earth conditions. For the process of chemical evolution on the early Earth, we may rather envision a

more continuous synthesis, in which small organic molecules, initially formed by volcanic action or lightning, reacted to give increasingly more complex structures (Fig. 1). Here, chemical transformations may have been driven by physical fluctuations, established for example by day-night, seasonal or wet–dry cycles. Such fluctuating parameters might include temperature, concentration, and pH, which have triggered selective precipitation and crystallization to purify and concentrate reaction intermediates (Fig. 2a).

Regarding the central nucleoside building blocks of life, we believe that the four canonical nucleosides were finally selected from a more diverse prebiotic nucleoside pool. These canonical bases today establish the sequence information. The synthesis of the canonical purine (A, G)[18] and pyrimidine (U, C)[16] RNA building blocks has been previously demonstrated in aqueous environments. It is questionable, however, if these multistep synthesis pathways are able to provide all four canonical bases at the same time, which fuels the development of new prebiotically plausible nucleoside formation reactions[17, 26, 27]. Recently, all four canonical nucleosides (A, G, U, C) were accessed in low yields via a one-pot procedure from pure formamide[28]. However, in order to establish a functional genetic system a number of non-canonical nucleosides is required as well that provide other functions related to folding and catalysis[29–34]. Since many of these non-canonical bases are present in all three domains of life, it is likely that they have been early on part of the abiotic chemical selection process[15]. We report here the discovery of a continuous synthesis pathway that enables the efficient production of

canonical and non-canonical purine bases in parallel. Our data show that formation of the many nucleosides needed to establish a functional genetic system is in fact an unavoidable event if we assume the presence of simple starting materials such as formic acid, acetic acid, sodium nitrite, malononitrile (**1**), amidinium compounds (**2a-d**), as well as transition metals like Ni or Fe. These simple compounds react in several successive wet–dry phases, leading to physical enrichment (I, II, and III) of reaction intermediates to finally give RNA building blocks. Wet–dry cycles have already been shown to be a plausible geological scenario especially for polymerization reactions[35, 36]. Our reported chemical scenario shows now that such a geological setup can also result in a diverse set of purine nucleosides by continuous synthesis (**6a-h**, Fig. 4). These nucleosides can be converted into the phosphorylated nucleotides based on recent advances in prebiotic phosphorylation reactions[37, 38]. So far, however, we are not yet able to include this step into our continuous synthesis.

Importantly, all here-reported non-canonical bases are known to exist in the three domains of life. Many are postulated components of the early genetic system of the last universal common ancestor (LUCA), suggesting that they were indeed present already at the onset of biological evolution[39, 40]. Based on the continuous synthesis pathway reported here, we hypothesize that the canonical and at least these non-canonical nucleosides could have formed side by side, dating the formation of the first non-canonical nucleosides back to the origin of chemical evolution around 4 billion years ago. As such, they could have been part of the chemical evolution process that established the putative RNA world[41]. Our chemistry invokes that methylated and thiomethylated nucleosides could particularly have been integral components of the first instructional (pre)-RNA molecules, likely to stabilize folded structures in order to accelerate catalytic processes[29–34]. We therefore propose that these nucleosides could be vestiges and molecular fossils of an early Earth, as it was suggested for cofactors[42].

## Methods

**1-methylguanidine (2a) salt of (hydroxyimino)malononitrile (3).** 1-methylguanidine (**2a**) hydrochloride salt (10.95 g, 100 mmol, 1 eq.) and malononitrile (**1**) (6.65 g, 100 mmol, 1 eq.) was dissolved in $H_2O$ (230 mL, containing 6 mL of AcOH) in a 500 mL beaker. A solution of $NaNO_2$ (7.00 g, 101 mmol, 1.01 eq., in 20 mL of $H_2O$) was slowly added at room temperature. After stirring at room temperature for 2 h the reaction mixture was kept at 45 °C in an oil bath for 3–4 days open to the air until the mixture was concentrated to about 100 mL. The reaction mixture was placed in a fridge overnight at 8–10 °C. The formed yellow crystals were filtered off to give the desired product (6.70 g, 40 mmol, 40%).

**Mp.:** 108 °C. **$^1$H-NMR** (400 MHz, DMSO-$d_6$) $\delta = 7.23$ (br m, 5H), 2.72 (s, 3H). **$^{13}$C-NMR** (101 MHz, DMSO-$d_6$) $\delta = 157.85, 119.50, 113.31, 107.18, 28.23$. **IR** (cm$^{-1}$): 3405 (m), 3351 (br, m), 3197 (m), 2977 (br, m), 2229 (s), 2218 (s) 1675 (s), 1635 (s), 1465 (w) 1428 (m), 1344 (s), 1269 (s) 1226 (s), 1172 (w), 1098 (m), 915 (m), 765 (m).

**Methylthioamidine (2b) salt of (hydroxyimino)malononitrile (3).** S-Methyl-isothiourea (**2b**) hemisulfate salt (27.8 g, 200 mmol, 1 eq.) and malononitrile (**1**) (13.3 g, 200 mmol, 1 eq.) was dissolved in $H_2O$ (460 mL, containing 12 mL of AcOH) in a 600 mL beaker. A solution of $NaNO_2$ (14.0 g, 202 mmol, 1.01 eq., in 40 mL of $H_2O$) was slowly added at room temperature. After stirring at room temperature for 2 h the reaction mixture was kept at 45 °C in an oil bath for 3–4 days open to the air until the mixture was concentrated to about 200 mL. The reaction mixture was placed in a fridge overnight at 8–10 °C. The formed yellow crystals were filtered off to give the desired product (16.7 g, 90 mmol, 45%).

**Mp.:** 126 °C. **$^1$H-NMR** (400 MHz, DMSO-$d_6$) $\delta$ 8.90 (s, 4H), 2.56 (s, 3H). **$^{13}$C-NMR** (101 MHz, DMSO-$d_6$) $\delta$ 171.62, 119.46, 113.29, 107.17, 13.71. **IR** (cm$^{-1}$): 3282(m), 3146 (br, m), 2742 (br, w), 2530 (w), 2222 (s), 2213 (s) 1698 (m), 1663 (s), 1643 (s) 1549 (m), 1450 (m), 1424 (s) 1375 (w), 1335 (s), 1269 (s), 1223 (s), 1180 (w), 1099 (m), 1076 (w), 982 (m), 970 (w), 960 (w), 897 (br, m), 801 (m), 736 (m).

**Guanidine (2c) salt of (hydroxyimino)malononitrile (3).** Guanidine (**2c**) hydrochloride salt (9.55 g, 100 mmol, 1 eq.) and malononitrile (**1**) (6.65 g, 100 mmol, 1 eq.) was dissolved in $H_2O$ (230 mL, containing 6 mL of AcOH) in a 500 mL beaker. A solution of $NaNO_2$ (7.00 g, 101 mmol, 1.01 eq., in 20 mL of $H_2O$) was

slowly added at room temperature. After stirring at room temperature for 2 h the reaction mixture was kept at 45 °C in an oil bath for 3–4 days open to the air until the mixture was concentrated to about 100 mL. The reaction mixture was placed in a fridge overnight at 8–10 °C. The formed yellow crystals were filtered off to give the desired product (8.20 g, 53 mmol, 53%).

**Mp.:** 159 °C. **$^1$H-NMR** (400 MHz, DMSO-$d_6$) $\delta = 6.91$ (s, 6H). **$^{13}$C-NMR** (101 MHz, DMSO-$d_6$) $\delta = 158.32, 119.52, 113.32, 107.18$. **IR** (cm$^{-1}$): 3473 (m), 3373 (m), 3172 (w), 3087 (w), 2815 (br, w), 2223 (s), 2217 (s), 1668 (m), 1641 (s), 1578 (w), 1552 (w), 1369 (w), 1343 (s), 1294 (w), 1264 (s), 1220 (s), 1140 (w), 975 (w), 792 (w), 757 (w).

**Acetamidine (2d) salt of (hydroxyimino)malononitrile (3).** Acetamidine (**2d**) hydrochloride salt (9.45 g, 100 mmol, 1 eq.) and malononitrile (**1**) (6.65 g, 100 mmol, 1 eq.) was dissolved in $H_2O$ (230 mL, containing 6 mL of AcOH) in a 500 mL beaker. A solution of $NaNO_2$ (7.00 g, 101 mmol, 1.01 eq., in 20 mL of $H_2O$) was slowly added at room temperature. After stirring at room temperature for 2 h the reaction mixture was kept at 45 °C in an oil bath for 3–4 days open to the air until the mixture was concentrated to about 100 mL. The reaction mixture was placed in a fridge overnight at 8–10 °C. The formed yellow crystals were filtered off to give the desired product (8.80 g, 60 mmol, 60%).

**Mp.:** 142 °C. **$^1$H-NMR** (400 MHz, DMSO-$d_6$) $\delta = 8.60$ (s, 4H), 2.11 (s, 3H). **$^{13}$C-NMR** (101 MHz, DMSO-$d_6$) $\delta = 168.11, 119.50, 113.31, 107.18, 18.72$. **IR** (cm$^{-1}$): 3282 (m), 3140 (br, m), 2781 (m), 2395 (w), 2236 (s), 2217 (s) 1708 (s), 1661 (s), 1592 (br, w) 1513 (m), 1373 (s), 1351 (s) 1259 (s), 1200 (s), 1191 (s), 1160 (w), 1125 (m), 969 (w), 907 (w), 883 (w), 857 (w), 790 (s), 691 (s), 684 (s).

**Synthesis of nitrosopyrimidine 4a-d from salts containing 2a-d and 3.** The reaction time of the following procedures for the formation of **4a-d** depend on the crystal size of the organic salts and the heating source. Usually conversions were done in a beaker open to the air in an oil bath to simulate hot soil. Large crystals sometimes already convert into nitroso-pyrimidines by a solid-state reaction without melting. Then much longer reaction times of up to 7 days are required because of unequally distributed temperature. Alternatively, the organic salt can be converted by heating in an oven where temperature is equally distributed within the sample.

**6-imino-1-methyl-5-nitroso-1,6-dihydropyrimidine-2,4-diamine (4a).** 1-methylguanidine (**2a**) salt of (hydroxyimino)malononitrile (**3**) (5.00 g, 29.5 mmol, 1 eq.) was heated slowly to its melting temperature of 108 °C and kept overnight. The compound melts suddenly but becomes a dark red solid again after leaving it overnight. The quantitative reaction mixture can be directly used for the next step without purification.

For analytical reasons a small batch of 1-methylguanidine (**2a**) salt of (hydroxyimino)malononitrile (**3**) (100 mg, 0.59 mmol, 1 eq.) was reacted as described above. The reaction mixture was suspended in water (2–3 mL) and the dark red solid was filtered off to give 6-imino-1-methyl-5-nitroso-1,6-dihydropyrimidine-2,4-diamine (59 mg, 0.35 mmol, 59%).

**$^1$H-NMR** (400 MHz, DMSO-$d_6$) $\delta = 11.44$ (s, 1H), 8.41 (s, 1H), 8.08 (br, 1H), 7.67 (br, 1H), 7.49 (s, 1H), 3.26 (s, 3H). **$^{13}$C-NMR** (101 MHz, DMSO-$d_6$) $\delta = 165.44, 157.64, 146.95, 137.22, 27.64$. **HRMS** (ESI + ): calc. for $[C_5H_9N_6O]^+$ 169.0832, found: 169.0832 $[M + H]^+$

**2-(methylthio)-5-nitrosopyrimidine-4,6-diamine (4b).** Methylthioamidine (**2b**) salt of (hydroxyimino)malononitrile (**3**) (5.00 g, 27 mmol, 1 eq.) was heated slowly to its melting temperature of 126 °C and kept overnight. Caution: if the product is heated too quickly above the melting temperature it decomposes with the release of MeSH! The compound melts suddenly but becomes a dark green solid again. The quantitative reaction mixture can be directly used for the next step without purification.

For analytical reasons a small batch of methylthioamidine (**2b**) salt of (hydroxyimino)malononitrile (**3**) (100 mg, 0.54 mmol, 1 eq.) was reacted as described above. The reaction mixture was suspended in water (2–3 mL) and the dark green solid was filtered off to give 2-(methylthio)-5-nitrosopyrimidine-4,6-diamine (64 mg, 0.35 mmol, 64%).

**$^1$H-NMR** (400 MHz, DMSO-$d_6$) $\delta$ 10.18 (d, $J = 4.2$ Hz, 1H), 9.00 (s, 1H), 8.42 (d, $J = 4.2$ Hz, 1H), 8.02 (s, 1H), 2.46 (s, 3H). **$^{13}$C-NMR** (101 MHz, DMSO-$d_6$) $\delta$ 179.05, 164.73, 146.22, 139.43, 14.08. **HRMS** (ESI+): calc. for $[C_5H_8N_5OS]^+$ 186.0444, found: 186.0444 $[M + H]^+$

**5-nitrosopyrimidine-2,4,6-triamine (4c).** Guanidine (**2c**) salt of (hydroxyimino)malononitrile (**3**) (5.00 g, 32 mmol, 1 eq.) was heated slowly to its melting temperature of 159 °C and kept overnight. The compound melts suddenly but becomes a red/pinkish solid again after leaving it overnight. The quantitative reaction mixture can be directly used for the next step without purification.

For analytical reasons a small batch of guanidine (**2c**) salt of (hydroxyimino)malononitrile (**3**) (100 mg, 0.65 mmol, 1 eq.) was reacted as described above. The reaction mixture was suspended in water (2–3 mL) and the solid was filtered off to give NMR clean 5-nitrosopyrimidine-2,4,6-triamine (85 mg, 0.55 mmol, 85%).

**¹H-NMR** (400 MHz, DMSO-$d_6$) $\delta$ = 10.26 (d, $J$ = 5.1 Hz, 1H), 8.15 (s, 1H), 7.75 (d, $J$ = 5.1 Hz, 1H), 7.35 (s, 1H), 7.19 (s, 2H). **¹³C-NMR** (101 MHz, DMSO-$d_6$) $\delta$ = 166.52, 165.32, 151.43, 138.04. **HRMS** (ESI+): calc. for $[C_4H_7N_6O]^+$ 155.0676, found: 155.0676 $[M + H]^+$

**2-methyl-5-nitrosopyrimidine-4,6-diamine (4d)**. Acetamidine (**2d**) salt of (hydroxyimino)malononitrile (**3**) (5.00 g, 32.5 mmol, 1 eq.) was heated slowly to its melting temperature of 142 °C and kept overnight. The compound melts suddenly but becomes a red/pinkish solid again after leaving it overnight. The quantitative reaction mixture can be directly used for the next step without purification.

For analytical reasons a small batch of acetamidine (**2d**) salt of (hydroxyimino) malononitrile (**3**) (100 mg, 0.65 mmol, 1 eq.) was reacted as described above. After suspension in $H_2O$ (2–3 mL) the solid is filtered off to give NMR clean 2-methyl-5-nitrosopyrimidine-4,6-diamine (71 mg, 0.46 mmol, 71%).

**¹H-NMR** (400 MHz, DMSO-$d_6$) $\delta$ = 10.04 (d, $J$ = 3.2 Hz, 1H), 8.97 (s, 1H), 8.36 (d, $J$ = 3.2 Hz, 1H), 7.96 (s, 1H), 2.20 (s, 3H). **¹³C-NMR** (101 MHz, DMSO-$d_6$) $\delta$ = 175.20, 166.38, 146.76, 139.94, 26.83. **HRMS** (ESI-): calc. for $[C_5H_6N_5O]^-$ 152.0578, found: 152.0578 $[M-H]^-$

**Data availability**. All data generated or analyzed during this study are presented in this article and its Supplementary Information File, or are available from the corresponding author upon reasonable request. X-ray crystallographic data were also deposited at the Cambridge Crystallographic Data Centre (CCDC) under the following CCDC deposition numbers: 1574226 for 1-methylguanidine (**2a**) salt of (hydroxyimino)malononitrile (**3**); 1574223 for methylthioamidine (**2b**) salt of (hydroxyimino)malononitrile (**3**); 1574225 for guanidine (**2c**) salt of (hydroxyimino)malononitrile (**3**); 1574224 for acetamidine (**2d**) salt of (hydroxyimino) malononitrile (**3**). These can be obtained free of charge from CCDC via www.ccdc. cam.ac.uk/data_request/cif.

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

## Acknowledgements

We thank the Deutsche Forschungsgemeinschaft for financial support through the research funding schemes SFB749, SFB1032, SPP1784, GRK2062, and the DFG Nor-malverfahrensprogram CA275-11/1. We thank the European Union Horizon 2020 program for funding the ERC Advanced project EPiR (741912). Further we thank the excellence cluster CiPS-M. We thank Florian Steinmann for synthetic help, Dr Markus Müller for further discussions and Dr Peter Mayer for providing X-ray structures.

## Author contributions

S.B. developed the chemistry for the nitrosopyrimidine route, helped designing the study, analyzed and interpreted results and helped writing the manuscript. C.S., H.O., A.C., T.A. and M.D. performed supportive chemistry. T.C. designed the study, supervised all work, interpreted data and wrote the manuscript.

## Additional information

**Competing interests:** The authors declare no competing financial interests.

