## [Peer Review File · Nature Communications]

Editorial Note: This manuscript has been previously reviewed at another journal that is not operating a transparent peer review scheme. This document only contains reviewer comments and rebuttal letters for versions considered at Nature Communications. Mentions of prior referee reports have been redacted.

Reviewers' comments:

Reviewer #1 (Remarks to the Author):

This paper presents novel and interesting results on the mechanisms of synthesis of canonical and non-canonical nucleosides in conditions relevant to the early Earth. This is very relevant to the issue of the origins of life. The general idea of driving reactions by wet-dry cycles and the fact that diverse mixtures of similar compounds are formed both seem important and plausible. This paper makes some significant steps forward in integrating the different steps of the reaction pathways.

Fig 4 is probably the most useful one in the paper. I would suggest that it should be moved to the beginning as Fig 1. Similarly, the sentence "The central assembly step involves...", which follows after Fig 4, is key to understanding the paper, and should be at the beginning. I was lost in the details until I reached this point in the paper.

I would encourage the authors to think about how to make these results as understandable as possible to readers without a strong chemistry background. I am struggling to interpret things which are probably fairly trivial for the authors. It is worth thinking about clarifying these things in order to increase the impact of the paper.

For example, a list of abbreviations and full names and chemical structures could be included in the supplementary information and pointed to clearly from the text. One abbreviation that troubled me particularly was 'FaPy', which is used in Fig 1 and Fig 2, but is not defined until page 6. In general, it takes me a long time to link the names of the compounds, which are sometimes used in the text and the figure captions, to the figures themselves, which contain the structures with no names.

Is there a reason why there are square brackets for one row of compounds in Fig 2. This step (with the NH₂) is not mentioned elsewhere - e.g. Fig 4 goes from nitroso- to formamido- compounds. In Fig 3 - do we need all the UV and MS results for every compound? What is the reader supposed to see from these? Would it be better to give just a few examples with a size that is big enough to read?

The word 'model' is used in the title and elsewhere. But I am assuming that the reaction scheme is fully tested, because experimental measurements and yields are given. In other words - the use of 'model' does not imply that the reaction scheme is proposed without experimental support. It would be worth emphasizing which parts have been tested.

Again for the non-chemist - why do we start from "(hydroxyimino)malononitrile and amidine molecules". Are these obvious components of the small-molecule prebiotic mixture?

The paper refers to wet-dry cycling. But are cycles really required, rather than just a single drying step? It is not clear to what extent the physical enrichment parts in the cartoons of Fig 2 have been tested.

Are the conditions in which nucleosides form compatible with polymerization? What about phosphate groups? Can the phosphate be added in the same reaction scheme? Depurination has been observed in experiments on formation of RNA oligomers via wet-dry cycles (e.g. Mungi and Rajamani *Life* 2015, 5(1), 65-84; doi:10.3390/life5010065). Are the purine nucleosides formed by this mechanism stable?

Reviewer #2 (Remarks to the Author):

Identifying prebiotically viable pathways to the canonical and non-canonical nucleosides/nucleotides remain a major challenge in the contemporary fields of prebiotic chemistry and the "origins of life". Here the authors report that wet/dry cycles, temperature fluctuations and physical separation steps (e.g., crystallization), facilitate reaction pathways toward such nucleosides when starting from simple building blocks (e.g., malononitrile, amidines, etc.). The beauty of the work is therefore reflected in the simple components and conditions employed, which can be viewed as prebiotically viable.

This is an important paper, which is worthy of publication in Nature Comm. Relatively minor issues that need to be addressed are articulated below:

1. Some of the sentences used by the authors are structured in a rather complex fashion and some are logically inaccurate. Examples:

Abstract:

"The question of how a complex multistep chemical synthesis of RNA building blocks was possible in such an environment remains unknown".

It is not the question that is "unknown", but rather the answer (or the pathways).

" Consequently, the today-found modified nucleosides"

Perhaps use: "Consequently, the contemporary modified nucleosides"

Page 9

"The central assembly step involves reaction of the nitroso-pyrimidines with formic acid in the presence of elementary metal like Ni or Fe to give formamido-pyrimidines, which if combined with ribose..."

Perhaps change to " ...which when combined with ribose..."

"Based on the here reported continuous model..."

Change to:

"Based on the continuous model reported here ..."

2. Compound 2 is mentioned in the text but is not shown numbered in any of the Figures (the structure appears in a deprotonated form in Figure 4 but then it is not numbered).

3. Figure 2. The formation path for 5b and 5c is somewhat confusing as it might suggest they are formed simultaneously (yielding 101%).

4. Figure 2. The authors use Earth's crust (bottom left). In their experiments they have employed pure metals, so while correct in principle, claiming the processes can be catalyzed by ("generic") earth crust is a bit of a stretch.

5. SI. Several typos. See compound 8, 14 (likely copy/paste errors).

6. SI. Nomenclature changes between compound 14 (no specific designation of absolute configuration) and compound 15 onward.

7. Figure S3. It would be useful to indicate which wavelength was used for monitoring the chromatograms (although it is somewhat confusing as it is listed as an LC/MS analysis).

Reviewer #3 (Remarks to the Author):

The manuscript "A continuous model for parallel origin of canonical and non-canonical nucleosides driven by wet-dry cycles" by Becker et al. is based on the assumption that natural cycles (in this case wet-dry cycles) might have established fluctuations generating complexity. The experimental model presented is very interesting, but the involvement of natural cycles in prebiotic affairs is not completely new. Literature precedents should be discussed more.

The experimental set up is well planned, well detailed, convincing.

In many instances statements are excessively blunt:

**Abstract (page 1/line 3): "These conditions MUST have allowed ..." . Yes, ... maybe.

** Abstract, last two lines. This conclusion is not logic. A chemical model is mixed with an evolutionary scenario. The data and the model described in this paper do not provide any cause-effect relationship between these domains. Nor the data nor the model justify considering modified nucleosides just as competitors and fossils.

** (2/11-13) "In fact, all known life is dependent on modified nucleosides". Life is dependent on many things, aminoacids, carboxylic acids, etc etc. Modified nucleosides are important, among many other things. Should soften.

** (2/15-17): "All reported chemical models so far rely on tightly controlled laboratory conditions and the isolation and purification of central reaction intermediates by sophisticated methods". Also this paper describes a multi-step procedure, requiring changing the reaction conditions and separation of the intermediates. One-pot synthesis of nucleosides without external intervention was reported (Saladino et al., Meteorite-catalyzed syntheses of nucleosides and of other prebiotic compounds from formamide under proton irradiation, Proc. Natl. Acad. Sci. USA, (2015) 112, E2746-E2755). This paper is not quoted.

** Considering certain purines as the canonical nucleosides, while all the rest (formamido-pyrimidines included) are considered as non-canonical competitors seems to be an extreme a posteriori point of view. This generates confusion, mixing chemistry with evolutionary biology. This is a key point in the logics of the paper, and needs initial clarification and additional discussion.

** What about "canonical" pyrimidines? RNA also requires U and C. Wouldn't this point be worth a line? If this is satisfactorily commented, I am in favor of acceptance. Otherwise I am not.

Minor points.

- The (hydroxyimino) malonitrile intermediate should be numbered in Figure 1
- Bond distances should be given in angstrom (not in pm)
- Page 4, line 5: remove the repeated point

Response to reviewer 1:

Comment 1: „Fig 4 is probably the most useful one in the paper. I would suggest that it should be moved to the beginning as Fig 1. Similarly, the sentence "The central assembly step involves...", which follows after Fig 4, is key to understanding the paper, and should be at the beginning. I was lost in the details until I reached this point in the paper.”

Response 1: *We agree that Fig. 4 gives a good overview for the synthetic pathway and we therefore moved it to the beginning and re-numbered all relevant compounds. We also incorporated the mentioned sentence “the central assembly step involves...” into the introduction for clarification of our results and defined abbreviations like nitrosoPy and FaPy, which we use throughout the text or figures.*

Comment 2: “I would encourage the authors to think about how to make these results as understandable as possible to readers without a strong chemistry background. [...] For example, a list of abbreviations and full names and chemical structures could be included in the supplementary information and pointed to clearly from the text. One abbreviation that troubled me particularly was 'FaPy', which is used in Fig 1 and Fig 2, but is not defined until page 6. In general, it takes me a long time to link the names of the compounds, which are sometimes used in the text and the figure captions, to the figures themselves, which contain the structures with no names.”

Response 2: *The new figure 1 contains now the numbers, names and structures of the molecules (nitrosopyrimidines and formamidopyrimidines). We also point more often to the relevant figures from the text to link numbers and/or names to the mentioned molecules. Abbreviations like FaPy should now be clearer (see response 1).*

Comment 3: “Is there a reason why there are square brackets for one row of compounds in Fig 2. This step (with the NH₂) is not mentioned elsewhere - e.g. Fig 4 goes from nitroso- to formamido-compounds.”

Response 3: *“The compounds in square brackets are non-isolated reaction intermediates. This is now defined in the figure caption. We also rewrote the sentence in the text that refers to these reaction intermediates: “This leads to reduction of the nitroso-pyrimidines 4 to aminopyrimidines as non-isolated reaction intermediates (Fig 3a, in square brackets), which are immediately formylated to give the water soluble formamidopyrimidines (FaPys) 5a-h in a one-pot reaction.”*

Comment 4: “In Fig 3 - do we need all the UV and MS results for every compound? What is the reader supposed to see from these? Would it be better to give just a few examples with a size that is big enough to read?”

Response 4: *Since all compounds behave differently (as reflected in the yields or side products) we prefer including data for all the discussed compounds. We agree that the figure is too small and should be displayed on a separate paper in landscape format. We will suggest this to the editors.*

Comment 5: “The word 'model' is used in the title and elsewhere. But I am assuming that the reaction scheme is fully tested, because experimental measurements and yields are given. In other words - the use of 'model' does not imply that the reaction scheme is proposed without experimental support. It would be worth emphasizing which parts have been tested.”

Response 5: *The chemical scheme is fully tested. We therefore changed the expression “continuous model” to “continuous synthesis”, which refers to the fact that all reaction intermediates are only enriched or purified by changes in physical parameters and are directly used for the next synthetic step. To clarify, we included two additional sentences in the introduction: “These fluctuations enable the direct enrichment or purification of all reaction intermediates that are directly used for the next*

synthetic steps. As such a continuous synthesis is established.” Accordingly we also changed the title to “Wet-dry cycles enable the parallel origin of canonical and non-canonical nucleosides by continuous synthesis” to remove the word “model”.

Comment 6: “Again for the non-chemist - why do we start from “(hydroxyimino)malononitrile and amidine molecules”. Are these obvious components of the small-molecule prebiotic mixture?”

Response 6: *We start from malononitrile which is converted (hydroxyimino)malononitrile in situ. We recognized that this was not fully clear and therefore numbered malononitrile as 1. Together with amidine molecules these starting materials are plausible prebiotic building blocks (see reviewer 2). We included a citation where both compounds have been used (Becker et al. Science 2016).*

Comment 7: “The paper refers to wet-dry cycling. But are cycles really required, rather than just a single drying step? It is not clear to what extent the physical enrichment parts in the cartoons of Fig 2 have been tested.”

Response 7: *The physical enrichment steps have been fully tested, which is necessary to establish the continuous synthesis (see response 5). We assume that the 3 physical enrichment steps (I, II and III) are following each other and therefore establish a sequence of wet-dry cycles, leading to continuous synthesis of RNA building blocks. We changed a sentence in the discussion accordingly to reflect this view: “These simple compounds react in several successive wet-dry phases, leading to physical enrichment (I, II and III) of reaction intermediates to finally give RNA building blocks.”*

Comment 8: “Are the conditions in which nucleosides form compatible with polymerization? What about phosphate groups? Can the phosphate be added in the same reaction scheme? Depurination has been observed in experiments on formation of RNA oligomers via wet-dry cycles (e.g. Mungi and Rajamani Life 2015, 5(1), 65-84; doi:10.3390/life5010065). Are the purine nucleosides formed by this mechanism stable?”

Response 8: *Phosphate groups can be added but this would probably need an additional wet-dry cycle. Literature is known to phosphorylate nucleosides (recent papers: ACIE, 2016, 55, 15816 or ACIE, 2016, 55, 13249). We added two sentences about phosphorylation in the discussion: “These nucleosides can be converted into the phosphorylated nucleotides based on recent advances in prebiotic phosphorylation reactions.^{38,39} So far, however, we are not yet able to include this step into our continuous synthesis.”*

We added a sentence in the discussion about wet-dry cycles in the context of polymerization reactions: “Wet-dry cycles have already been shown to be a plausible geological scenario especially for polymerization reactions.” The mentioned conditions for polymerization (Mungi and Rajamani, Life 2015) most likely would be problematic for all kinds of nucleosides since pH 2 or below together with high temperatures is required. The conditions will hydrolyse the glycosidic C-N bonds in the presence of water very quickly. Nucleosides are most stable under slightly acidic, neutral or slightly basic conditions.

Response to reviewer 2:

Comment 1: ""The question of how a complex multistep chemical synthesis of RNA building blocks was possible in such an environment remains unknown". It is not the question that is "unknown", but rather the answer (or the pathways)."

Response 1: *We changed "unknown" to "unanswered"*

Comment 2: "" Consequently, the today-found modified nucleosides" Perhaps use: Consequently, the contemporary modified nucleosides"

Response 2: *We changed that sentence completely (see Reviewer 3, Response 2B).*

Comment 3: ""The central assembly step involves reaction of the nitroso-pyrimidines with formic acid in the presence of elementary metal like Ni or Fe to give formamido-pyrimidines, which if combined with ribose..." Perhaps change to " ...which when combined with ribose..."

Response 3: *We moved this sentence into the introduction (as suggested by reviewer 1) and rewrote it as suggested by reviewer 2: "[...] When combined with ribose, the FaPy compounds deliver a set of purine nucleosides."*

Comment 4: ""Based on the here reported continuous model..." Change to: "Based on the continuous model reported here ..."

Response 4: *We changed that sentence accordingly: "Based on the continuous synthesis pathway report here, ..."*

Comment 5: "Compound 2 is mentioned in the text but is not shown numbered in any of the Figures (the structure appears in a deprotonated form in Figure 4 but then it is not numbered)."

Response 5: *Since we moved original Fig. 4 to the beginning as Fig. 1 we re-numbered the structures. The original compound 2 is now changed to compound number 3 and numbering occurred throughout the manuscript.*

Comment 6: "Figure 2. The formation path for 5b and 5c is somewhat confusing as it might suggest they are formed simultaneously (yielding 101%)."

Response 6: *We agree the mentioned arrows were misleading in the figure (which is now figure 3). The figure includes now two separate arrows, one for 5b and one for 5c, to indicate separate transformations (and yields).*

Comment 7: "Figure 2. The authors use Earth's crust (bottom left). In their experiments they have employed pure metals, so while correct in principle, claiming the processes can be catalyzed by ("generic") earth crust is a bit of a stretch."

Response 7: *We removed the term "Earth's crust" and replaced it simply by reduction, since the cartoon should illustrate a reductive environment, where the nitrosopyrimidines get reduced.*

Comment 8: "SI. Several typos. See compound 8, 14 (likely copy/paste errors)."

Response 8: *We went over the SI again and corrected found typos.*

Comment 9: "SI. Nomenclature changes between compound 14 (no specific designation of absolute configuration) and compound 15 onward."

Response 9: *We corrected accordingly.*

Comment 10: “Figure S3. It would be useful to indicate which wavelength was used for monitoring the chromatograms (although it is somewhat confusing as it is listed as an LC/MS analysis).”

Response 10: *We included the wavelength in the figure.*

Response to reviewer 3:

Comment 1: “The experimental model presented is very interesting, but the involvement of natural cycles in prebiotic affairs is not completely new. Literature precedents should be discussed more.”

Response 1: *We cited literature in the discussion and added a sentence: “Wet-dry cycles have already been shown to be a plausible geological scenario especially for polymerization reactions.”^{36,37}*

Comment 2: “In many instances statements are excessively blunt:”

- A) “Abstract (page 1/line 3): “These conditions MUST have allowed ...” . Yes, ... maybe.”
- B) “Abstract, last two lines. This conclusion is not logic. A chemical model is mixed with an evolutionary scenario. The data and the model described in this paper do not provide any cause-effect relationship between these domains. Nor the data nor the model justify considering modified nucleosides just as competitors and fossils.”
- C) “(2/11-13) “In fact, all known life is dependent on modified nucleosides”. Life is dependent on many things, aminoacids, carboxylic acids, etc etc. Modified nucleosides are important, among many other things. Should soften.”
- D) “(2/15-17): “All reported chemical models so far rely on tightly controlled laboratory conditions and the isolation and purification of central reaction intermediates by sophisticated methods”. Also this paper describes a multi-step procedure, requiring changing the reaction conditions and separation of the intermediates. One-pot synthesis of nucleosides without external intervention was reported (Saladino et al., Meteorite-catalyzed syntheses of nucleosides and of other prebiotic compounds from formamide under proton irradiation, Proc. Natl. Acad. Sci. USA, (2015) 112, E2746-E2755). This paper is not quoted.”
- E) Considering certain purines as the canonical nucleosides, while all the rest (formamido-pyrimidines included) are considered as non-canonical competitors seems to be an extreme a posteriori point of view. This generates confusion, mixing chemistry with evolutionary biology. This is a key point in the logics of the paper, and needs initial clarification and additional discussion.
- F) “What about “canonical” pyrimidines? RNA also requires U and C. Wouldn't this point be worth a line?”

Response 2:

- A) *We changed “must have allowed...” to “might have allowed...”*
- B) *We modified the sentence: “The data show that modified nucleosides were potentially formed as competitor molecules. They could in this sense be considered as molecular fossils.” To clarify further we included 2-3 sentences in the introduction how RNA might have evolved: “We need to consider, however, that an early genetic polymer might have been structurally different from contemporary RNA. This involves differences regarding the sugar configuration (eg. pyranosyl RNA) or the presence of other nucleobases.^{10,11} Selection pressure led in this scenario to the chemical evolution of RNA.” According to this scenario, different nucleosides available on early Earth can be considered as “competitors” and the ones that survived selection pressure can be viewed as “molecular fossils”*

- C) We agree that life depends on many chemical components therefore we defined more closely: “In fact, the genetic system of all known life is dependent on modified nucleosides.” Even if we would add “...among other components” is there any gain in information? Our sentence does not exclude other components, which are certainly present as well (like enzymes), but they are irrelevant for the topic of the paper.
- D) Again we defined more closely (“All reported multi-step chemical models...”) to exclude one-pot synthetic procedures like Saladino et al., Proc. Natl. Acad. Sci. USA, (2015) 112, E2746-E2755, which we cited (“Recently, all four canonical nucleosides (A, G, U, C) were accessed in low yields via a one-pot procedure from pure formamide.²⁹”).
- E) We define canonical nucleosides as the molecules responsible for establishing the sequence information (A, G, U, C) and non-canonical nucleosides as all other nucleosides found in RNA today that provide different functions. Canonical and non-canonical are established definitions. We agree that FaPys are neither canonical nor non-canonical, since they are not present in RNA. Therefore we removed the terms “non-canonical FaPys” or “canonical FaPys”.
- F) We added several sentences about prebiotic synthesis of nucleosides in general also discussing formation of U and C and cited most recent literature (Proc. Natl. Acad. Sci. 2017, **114**, 11315 and Scientific Reports 2017, **7**, 14709): “Regarding the central nucleoside building blocks of life, we believe that the four canonical nucleosides were finally selected from a more diverse prebiotic nucleoside pool. These canonical bases today establish the sequence information. The synthesis of the canonical purine (A, G)¹⁵ and pyrimidine (U, C)¹³ RNA building blocks has been previously demonstrated in aqueous environments. It is questionable, however, if these multi-step synthesis pathways are able to provide all four canonical bases at the same time, which fuels the development of new prebiotically plausible nucleoside formation reactions.^{14,27,28}”

Comment 3: “The (hydroxyimino) malonitrile intermediate should be numbered in Figure 1”

Response 3: We numbered the mentioned compound accordingly.

Comment 4: “Bond distances should be given in angstrom (not in pm)”

Response 4: We now give the distance in angstrom.

Comment 5: “Page 4, line 5: remove the repeated point”

Response 5: We changed the text accordingly.

Reviewers' Comments:

Reviewer #1:

Remarks to the Author:

The responses to my previous questions are very clear. I would recommend publication with no further change.

Reviewer #3:

Remarks to the Author:

The revised version of the ms NCOMMS-17-26411A answers the comments and queries. In my opinion it can now be published.